# Are Central Banks' Monetary Policies the Future of Housing Affordability Solutions

Chung Yim Yiu

Department of Property, University of Auckland Business School, Auckland 1010, New Zealand;
edward.yiu@auckland.ac.nz

**Abstract:** Housing affordability is one of the major social problems in many countries, with some advocates urging governments to provide more accessible mortgages to facilitate more homeownership. However, in recent decades more and more evidence has shown that unaffordable housing is the consequence of monetary policy. Most of the previous empirical studies have been based on econometric analyses, which make it hard to eliminate potential endogeneity biases. This cross-country study exploited the two global interest rate shocks as quasi-experiments to test the impacts and causality of monetary policy (taking real interest rates as a proxy) on house prices. Global central banks' synchronized reduction in interest rates after the outbreak of the COVID-19 pandemic in 2020 and then the global synchronized increase in interest rates after the global inflation crisis in 2022 provided both a treatment and a treatment reversal to test the monetary policy hypothesis. The stylized facts vividly reveal the negative association between interest rate changes and house price changes in many countries. This study further conducted a ten-country panel regression analysis to test the hypothesis. The results confirmed that, after controlling for GDP growth and unemployment factors, the change in real interest rate imposed a negative effect on house price growth rates. The key practical implication of this study pinpoints the mal-prescription of harnessing monetary policy to solve housing affordability issues, as it can distort housing market dynamics.

**Keywords:** monetary policy; real interest rate; COVID-19; global interest rate shocks; global house prices

## 1. Introduction

With the dramatic escalation in house prices much faster than household incomes in many countries in the past decades, housing affordability has become one of the major social problems globally. Numerous research studies have been conducted on the adverse impacts of housing affordability crises on households and society at large [1–3], especially on young people [4,5]. It has been put on the top priority agenda of policy makers in many countries, but almost all of them have focused on increasing land and social housing supply or deregulating urban planning restrictions to facilitate more private housing supply [6–8]. Unfortunately, after many years' and many countries' efforts toward increasing housing supply, the global housing affordability crisis does not seem to be resolved, but rather has been heightened to alarming proportions [9]. For example, a general deteriorating trend was found in the global housing affordability crisis based on a sample of 200 cities.

Some empirical studies, such as that of Fingleton (2008) [10], have found that simply increasing supply may not be sufficient to reduce house prices as demand factors play a role. Borgersen (2022) [11] also found that supply of social housing caused higher house prices and crowded out those who were not included in the social housing schemes. Thus, Raisner and Bracken (2022) [12] of the World Economic Forum contended that providing 'more mortgages for homebuyers', among others, is the solution to the global housing affordability problem in the future. Causa, Woloszko and Leite (2020) [13] agreed that 'access to mortgage debt for young household is likely to be one key driver of homeownership for this group, given their relatively low current wealth and income.' The literature has also confirmed

that young households are more sensitive to mortgage policy [14]. Interestingly, a relaxed mortgage policy, such as a lower interest rate or a higher loan-to-value ratio, was found to increase house prices, which rendered housing more unaffordable [15].

Ryan-Collins (2019) [16], for instance, raised a contention by asking 'why can't you afford a home?' in his book title. In contrast to the (insufficient) housing supply hypothesis, he put forward a monetary policy hypothesis and argued that it was the 'unlimited credit and money flows into an inherent finite supply of property that causes rising house prices.'

Mian and Sufi (2011) [17] also found empirical evidence on the association between house prices and household borrowing. Numerous pieces of empirical evidence have shown an association between monetary policy, especially interest rate, and house prices [18–27].

However, most of these previous empirical studies on this hypothesis have been based on econometric methods, which are subject to endogeneity biases [28,29], such as omitted variable bias, measurement errors and reverse causality [30]. Some studies have applied instrumental variables (IVs) to deal with endogeneity biases, but these have been found to be highly limited [31,32]. IVs are also not appropriate for resolving reverse causality bias.

This study focuses on dealing with the reverse causality bias. For example, the causal inference in a panel analysis of monetary policy impacts on house prices can be threatened by reverse causality [33]. The solution to reverse causality, as raised by Kenny (1979) [34], is to establish the temporal precedence of the independent variable to the dependent variable, among other conditions. One of the best solutions to establish temporal precedence is to use experimental trials which can apply interventions to a treatment group and not to a control group [33]. It is hard to conduct real experiments using randomized controlled trials in housing markets, but quasi-experiments on policy changes can be used. This study exploits the global interest rate shocks in the most recent three years as quasi-experiments to eliminate the reverse causality bias and test the monetary policy hypothesis.

After the outbreak of the COVID-19 pandemic in 2020, the synchronized actions of cutting interest rates by central bankers provides the first global interest rate shock as the treatment (intervention) to test the monetary policy hypothesis using a quasi-experimental approach. Since the shock was a response of central bankers to deal with the recessionary risks caused by the pandemic, it is an exogenous event to the housing markets.

Since February 2022, the outbreak of the Russian–Ukraine war has led the world into a global inflation crisis, triggering a series of global interest rate hikes at different paces and magnitudes. As the global inflation has been largely caused by the surges in fuel prices (oil and natural gas) and food prices due to the war and the supply chain disruptions, the interest rate hikes are also exogenous to the housing market situation.

These two global interest rate shocks provide both a treatment and a treatment reversal [35] to test the cause-and-effect relationship between monetary policy and house prices, which is the novelty of this study. As there are some countries that did not follow the interest rate shocks, they serve as controls for the quasi-experiments. This paper aims to examine the impact and causality of the global interest rate shocks on the global housing market, by empirically studying a ten-country panel regression analysis based on OECD (2022a) [36] data.

### 1.1. From Interest Rate Cuts to Interest Rate Hikes

During the COVID-19 period, many cities were locked down and the global supply chains were seriously disrupted, leading to a global recession. With economic activity grinding to a halt globally, central banks 'implemented unparalleled monetary policies, ultimately lowering their policy rates to historically depressed levels, even below zero' [37]. In 2020, after the outbreak of the pandemic, there were 207 interest rate cuts by central banks [38]. Just in 2020 Q1, there were 73 interest rate cuts by 54 central banks to support the economies and some were even cut to historic lows. Some central banks also relaxed liquidity rules on banks, such as the removal of the loan-to-value ratio restrictions on new mortgages in New Zealand. The global financial markets were flooded with liquidity in this period.

However, in the wake of a rapid deterioration in inflation, especially after the outbreak of the Russian–Ukraine war since February 2022, many central banks started increasing interest rates and turned the accommodative monetary policy into a contractionary one. In 2022 Q2, 55 central banks around the world turned around to raise interest rates 62 times, each time not less than 50 basis points [39]. In the first three quarters of 2022, the world recorded a total of 8190 basis points in central bank interest rate hikes, and just counting the G10 countries alone, their central banks raised interest rates by a total of 1850 basis points [40].

Taking the following ten countries as examples, viz. Australia (AUS), Canada (CAN), Germany (DEU), France (FRA), Great Britain (GBR), Italy (ITA), Japan (JAP), South Korea (KOR), New Zealand (NZL), and the United States of America (USA), as shown in Figure 1, most of their short-term interest rates started falling rapidly in 2020 Q2, and some of them, such as those in AUS, CAN, GBR, NZL and the USA, were cut to almost zero rates. ITA and JPN were the exceptions, as their short-term interest rates had been close to or below zero. Then, KOR and NZL raised their interest rates two-quarters earlier than others, but in general, most of them showed a rebound in short-term interest rates no later than 2022 Q2, except ITA and JPN. The pattern clearly shows U-shaped (downward and then upward) shocks in global interest rates within a short period, with one in 2020 and the other in 2022.

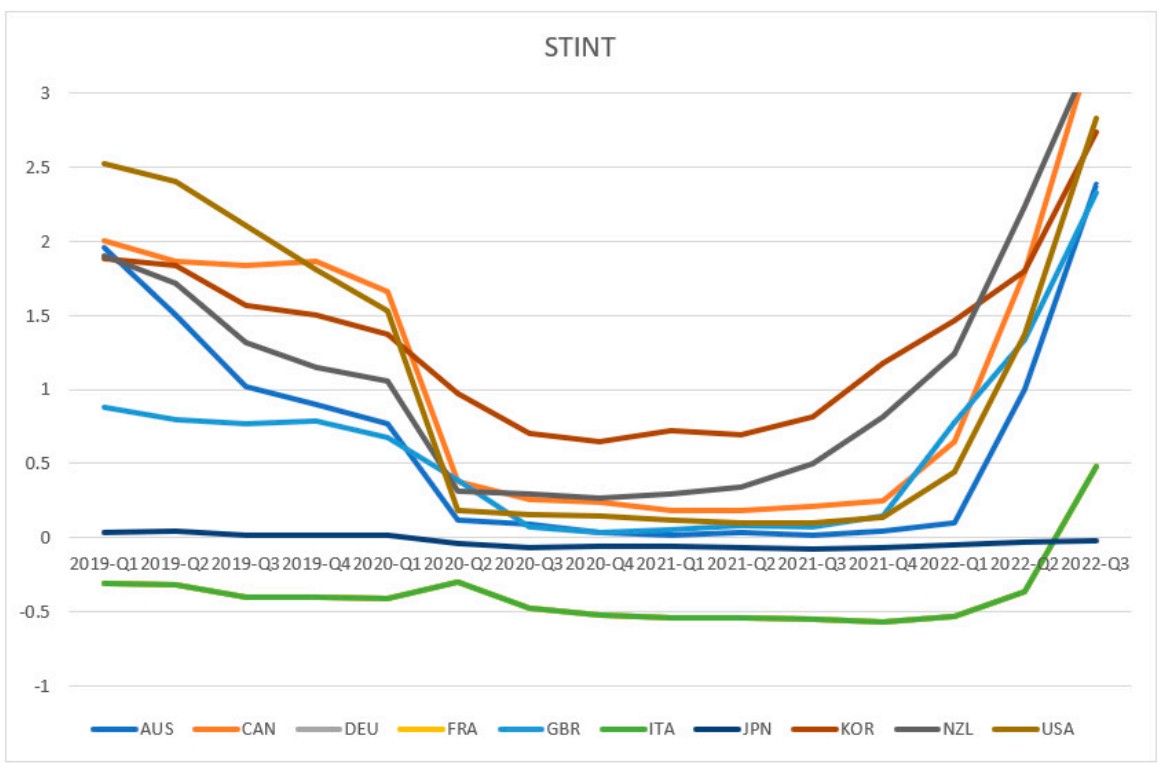

**Figure 1.** Short-term interest rates of ten countries from 2019 Q1 to 2022 Q3. Source: data from OECD (2022a) [36]. Legend: Australia (AUS), Canada (CAN), Germany (DEU), France (FRA), Great Britain (GBR), Italy (ITA), Japan (JAP), South Korea (KOR), New Zealand (NZL), and the United States of America (USA).

*1.2. From House Price Surges to House Price Falls*

Corresponding to the two recent global interest rate shocks, the house prices of many countries showed strong surges in 2021 and then fell in 2022. For example, the International House Prices Database reported that 24 out of 25 sampled countries recorded both annual and quarterly growth in real house prices in 2021 Q3 [41]. Some countries (New Zealand and Sweden) even recorded unprecedented highs in the annual growth rates of house prices (25.8% and 17.8%). However, in 2022 Q2 only 10 out of the 25 sampled countries

continued to have quarterly growth. Based on Knight Frank's (2021-2022) Global House Price Index, Figure 2 shows the annual growth rates of the house prices of the ten countries. Most of them indicate either a reducing rate of increase or even a negative figure in the annual growth rate in 2022. KOR, for example, showed a plummet from the peak of rising at 26.4% in 2021 Q3 to a fall of 7.5% in 2022 Q3. Similarly, in NZL the annual growth rate of house prices fell from a peak of 25.9% in 2021 Q2 to a negative 2.0% in 2022 Q3. ITA and JPN, on the other hand, kept steady growing trends.

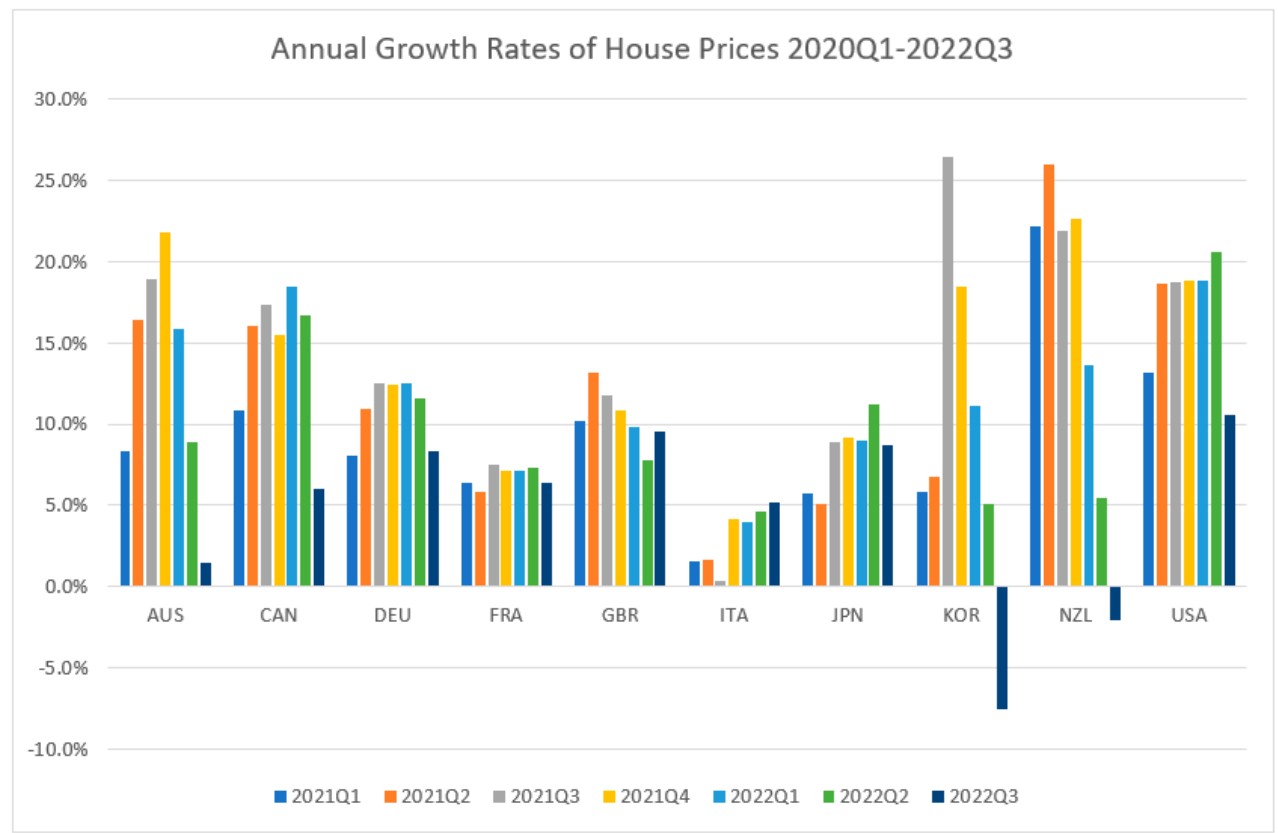

**Figure 2.** Annual growth rates of house prices of ten countries from 2020 Q1 to 2022 Q3. Source: data from Knight Frank (2021–2022) [42]. Legend: Australia (AUS), Canada (CAN), Germany (DEU), France (FRA), Great Britain (GBR), Italy (ITA), Japan (JAP), South Korea (KOR), New Zealand (NZL), and the United States of America (USA).

Since these two quasi-experiments with global interest rate shocks were carried out in different countries, it helps control for the peculiarities of individual countries, such as differences in planning regulations, land and housing supply constraints, etc. More importantly, as the interest rate shocks happened almost at the same time (synchronized), it helps eliminate omitted variable bias due to some unobservable temporal factors that affected the responses of the housing markets in different countries at different times.

This paper is organized as follows. Section 2 discusses the study gap based on a literature review. Section 3 presents the stylized facts with details of the data and the research design. Section 4 reports the empirical tests' results. Section 5 discusses the findings and implications, and Section 6 is the Conclusion.

## 2. Literature Review

There have already been many theoretical and empirical studies on the impacts of monetary policy on house prices, but many economists and politicians still disagree about the role that monetary policies, such as interest rates policy, play in causing house price booms and busts, probably because almost all the previous studies have not been based



on experiments, and monetary policy can include many tools. For example, Kuttner (2013: 2) [43] reviewed the standard economic theories (such as those in [44,45]) which explain why interest rate changes house values (or the value of any long-lived asset), and contended that interest rate alone could not explain the magnitude of house price changes, and instead it was the overall monetary policy, such as a loosening of credit conditions, plausibly caused by financial innovation (e.g. securitization), and a relaxation of lending standards that caused asset price bubbles. Taylor (2007, 2009) [46,47] also found that the asset price bubbles causing the US subprime crisis were created by an overly expansionary monetary policy, which included both the quantitative easing, ultra-low interest rates, etc. Yin, Su and Tao (2020) [48] found an association between monetary policy and housing price changes in Mainland China but they used M2 money supply as the proxy. Chadwick and Nahavandi (2022) [49] and Brunton and Jacob (2022) [50] also confirmed that monetary policy had impacts on house prices in New Zealand, where the policy included the removal of mortgage loan restrictions, ultra-low interest rates, etc. However, interest rate data are probably the only available leading data globally to represent monetary policy changes. It explains why real interest rates are still commonly used in empirical studies. Jordà, Schularick, and Taylor (2015) [23] also established the link between interest rates, mortgage lending, and house prices by exploiting the trilemma of international finance. This link supports the use of real interest rate as a proxy for monetary policy in studying house prices, as it is strongly associated with mortgage lending [51].

John Williams (2016) [52], the President and CEO of the Federal Reserve Bank of San Francisco, admitted that monetary policy is a common tool of central banks to foster financial stability, especially on house price stability, and contended that real estate finance has become a key source of risk for financial stability due to the devastating effects of a debt-fuelled housing crash. He further showed the significant and persistent effects of monetary policy on real house prices by means of a sample of 17 countries over the past 140 years.

However, there are a few studies that did not find a significant association between interest rate and house prices [53,54] and some studies have queried a reverse causality bias in the previous studies as they found a bidirectional relationship between mortgage policy and house prices [19]. In other words, the monetary policy hypothesis has not been confirmed, as most of the previous studies either have not controlled endogeneity issues among variables or have controlled them by means of econometric methods which are subject to limitations. Correlation does not necessarily imply causation [55], as causation requires counterfactual dependence [56] and temporal precedence [34]. This paper therefore adopts a quasi-experimental approach by using the monetary policies during the pandemic period as the interventions (both a treatment and a treatment reversal), which is a better way to test causality relationship, by means of the temporal precedence of the exogenous shocks and the counterfactual dependence of the treatment reversal.

## 3. Data and Research Design

### 3.1. Data

The quarterly house price data of ten countries were collected from the OECD (2022a) data platform. The ten countries are Australia (AUS), Canada (CAN), Germany (DEU), France (FRA), Great Britain (GBR), Italy (ITA), Japan (JAP), South Korea (KOR), New Zealand (NZL), and the United States of America (USA). They cover four continents, including two countries in North America, four countries in Europe, two countries in Asia, and two countries in Oceania, which could avoid a regional bias. These markets are also more transparent, with uniform data format and availability from the OECD data. Their market-oriented capitalism systems could also minimize institutional effects. As some of the house price series were not yet updated to 2022Q3, the missing data are estimated by using the annual growth rates provided by the Knight Frank Global House Price Indices [42].

Similarly, the macroeconomic data series, including the quarterly GDP growth rates, unemployment rates, nominal short- and long-term interest rates, and inflation rates of each country were collected from the OECD (2022a) [36]. These variables were included to test the top three macroeconomic determinants of house prices, viz. the economic factor [21]), the employment factor [57] and the monetary policy factor. Several robustness tests were also conducted by using long-term interest rates and cross-section weights to test their sensitivities to the terms of interest rates and the cross-country heteroscedasticity. Table 1 presents the descriptive statistics, and Table 2 reports the stationarities of the variables by using the Levin, Lin and Chu test and the Im, Peasaran and Shin test. They show that all the series are stationary in their first differences.

**Table 1.** Descriptive Statistics of Variables for 2015 Q1–2022 Q3. Sources: OECD (2022a) [36].

| Variable | Country | Mean | Standard Deviation | Minimum | Maximum |
|---|---|---|---|---|---|
| ΔHPI, House Price Index Quarter-on-Quarter Change | AUS | 0.013 | 0.025 | –0.031 | 0.025 |
| | CAN | 0.020 | 0.019 | –0.032 | 0.056 |
| | DEU | 0.018 | 0.007 | 0.008 | 0.034 |
| | FRA | 0.010 | 0.006 | –0.004 | 0.022 |
| | GBR | 0.014 | 0.011 | –0.003 | 0.039 |
| | ITA | 0.003 | 0.009 | –0.021 | 0.019 |
| | JPN | 0.008 | 0.010 | –0.007 | 0.032 |
| | KOR | 0.004 | 0.021 | –0.100 | 0.023 |
| | NZL | 0.022 | 0.026 | –0.027 | 0.081 |
| | USA | 0.020 | 0.015 | –0.020 | 0.049 |
| RSIR, Real Short-term Interest Rate (%) | AUS | –0.796 | 1.847 | –5.141 | 1.125 |
| | CAN | –1.152 | 1.685 | –5.755 | 0.386 |
| | DEU | –2.217 | 2.095 | –7.996 | 0.097 |
| | FRA | –1.600 | 1.423 | –5.647 | 0.287 |
| | GBR | –1.622 | 1.749 | –6.563 | 0.270 |
| | ITA | –1.583 | 2.071 | –7.908 | 0.294 |
| | JPN | –0.466 | 0.841 | –2.890 | 0.778 |
| | KOR | –0.121 | 1.371 | –3.607 | 1.525 |
| | NZL | –0.434 | 2.447 | –5.686 | 3.392 |
| | USA | –1.514 | 2.420 | –7.523 | 0.875 |
| ΔGDP, Gross Domestic Product Quarter-on-Quarter Change (%) | AUS | 0.615 | 1.768 | –6.758 | 3.855 |
| | CAN | 0.415 | 2.716 | –10.928 | 9.018 |
| | DEU | 0.291 | 2.494 | –9.481 | 9.005 |
| | FRA | 0.376 | 4.327 | –13.503 | 18.280 |
| | GBR | 0.425 | 5.083 | –20.991 | 16.609 |
| | ITA | 0.277 | 3.683 | –12.101 | 14.450 |
| | JPN | 0.124 | 1.946 | –7.952 | 5.575 |
| | KOR | 0.635 | 0.950 | –3.027 | 2.348 |
| | NZL | 0.747 | 3.376 | –10.394 | 13.675 |
| | USA | 0.529 | 2.192 | –8.484 | 7.854 |
| UNE, Unemployment Rate (%) | AUS | 5.450 | 0.809 | 3.476 | 7.089 |
| | CAN | 6.845 | 1.553 | 5.067 | 12.867 |
| | DEU | 3.555 | 0.465 | 2.900 | 4.500 |
| | FRA | 8.865 | 1.027 | 7.267 | 10.433 |
| | GBR | 4.435 | 0.574 | 3.600 | 5.600 |
| | ITA | 10.394 | 1.268 | 7.933 | 12.533 |
| | JPN | 2.794 | 0.340 | 2.300 | 3.500 |
| | KOR | 3.661 | 0.363 | 2.733 | 4.333 |
| | NZL | 4.465 | 0.729 | 3.200 | 5.700 |
| | USA | 4.938 | 1.873 | 3.567 | 12.967 |
| Period | | 2015 Q1–2022 Q3 | | | |
| Number of Observations | | 310 Obs (31 periods × 10 countries) | | | |

**Table 2.** Unit Root Tests of Variables, 2015 Q1–2022 Q3.

| Variable | Level | | First-Difference | |
|---|---|---|---|---|
| | Levin, Lin & Chu t* | Im, Peasaran and Shin W-stat | Levin, Lin & Chu t* | Im, Peasaran and Shin W-stat |
| log(HPI), log House Price Index and dlog (HPI), House Price Quarter-on-Quarter Change (%) | 6.96 | 7.92 | −1.82 ** | −4.40 *** |
| RSIR, Real Short-term Interest Rate (%) | 4.61 | 1.07 | −5.60 *** | −7.68 *** |
| RLIR, Real Long-term Interest Rate (%) | 6.10 | 6.20 | −7.13 *** | −6.58 *** |
| dlog(GDP), Gross Domestic Product Quarter-on-Quarter Change (%) | - | - | −23.74 *** | −19.51 *** |
| UNE, Unemployment Rate (%) and d(UNE) | −0.24 | −0.63 | −13.24 *** | −13.03 *** |

Notes: figures are statistics; ***, ** and * represent *p*-values ≤ 0.01, 0.05 and 0.10, respectively. The panel unit root tests used Newey-West automatic bandwidth selection and Quadratic Spectral kernel, with automatic lag length selection based on SIC: 0 to 5, and automatic selection of maximum lags.

*3.2. Research Design*

Equation (1) shows the panel regression model used to test the monetary policy hypothesis.

$$dlog(HPI_{i,t}) = \beta_1 d(RSIR_{i,t}) + \beta_2 dlog(GDP_{i,t}) + \beta_3 d(UNE_{i,t}) + \alpha_i + \gamma dlog(HPI_{i,t-1}) + \varepsilon_{i,t} \tag{1}$$

where the house price index ($HPI_{i,t}$), real short-term interest rate ($RSIR_{i,t}$), gross domestic product ($GDP_{i,t}$), and unemployment rate ($UNE_{i,t}$) of country $i$ at time $t$ are included. $\alpha_i$ controls cross-country fixed effects, and $\gamma dlog(HPI_{i,t-1})$ caters to the autoregressive effect of house price changes in a one-quarter lag (AR (1)). $\beta_1$, $\beta_2$, $\beta_3$, $\gamma$ are coefficients to be estimated. $\varepsilon_{i,t}$ is the error term.

A cross-country panel study, however, cannot sort out endogeneity biases. This study therefore made use of the hikes and falls in the global interest rates during the COVID-19 period as the treatment and the treatment reversal, respectively, of the quasi-experiments to study the monetary policy hypothesis. As stylized facts, Figure 3 shows the house price indices of the ten countries from 2015 Q1 to 2022 Q3 (2015 = 100). All showed positive growth rates in 2020 and 2021, with the peaks of NZL, CAN and the USA, respectively, ranked as the top three. However, five out of the ten house price indices started to decrease in 2022. The house price indices of AUS and NZL fell the earliest, with peaks in 2021 Q4, whereas those of CAN, the USA, and KOR fell later, with peaks in 2022 Q2. The house prices of the other five countries still showed increases in 2022 Q3, including those of ITA and JPN.

The pandemic has caused a global recession in 2020 and a rebound in 2021. Figures 4 and 5 show the GDP quarterly growth rates and the unemployment rates from 2015 Q1 to 2022 Q3 for the ten countries.

The quarterly growth rates of the GDPs of many countries were negative in 2020 and positive in 2021; similarly, unemployment rates were high in 2020, especially in the USA and CAN, and then became lower since 2021. These could not be the reasons for the house price rises in 2020–2021 and the falls in 2022. These special scenarios provide a strong test of the monetary policy hypothesis as other macroeconomic factors, such as GDP growth rates and unemployment rates, imposed opposite effects against the central banks' contrarian

efforts in adjusting their monetary policies to tackle the recessionary situations in 2020 and the inflationary ones in 2022.

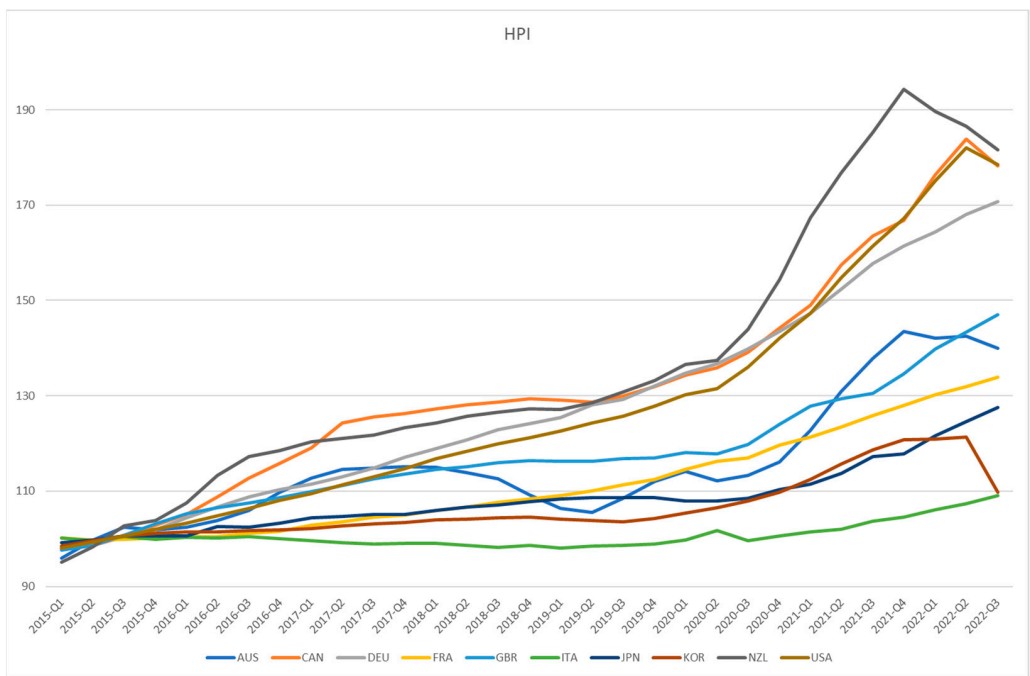

**Figure 3.** House price indices of ten countries from 2015 Q1 to 2022 Q3. Source: data from OECD (2022a) [36]. Legend: Australia (AUS), Canada (CAN), Germany (DEU), France (FRA), Great Britain (GBR), Italy (ITA), Japan (JAP), South Korea (KOR), New Zealand (NZL), and the United States of America (USA).

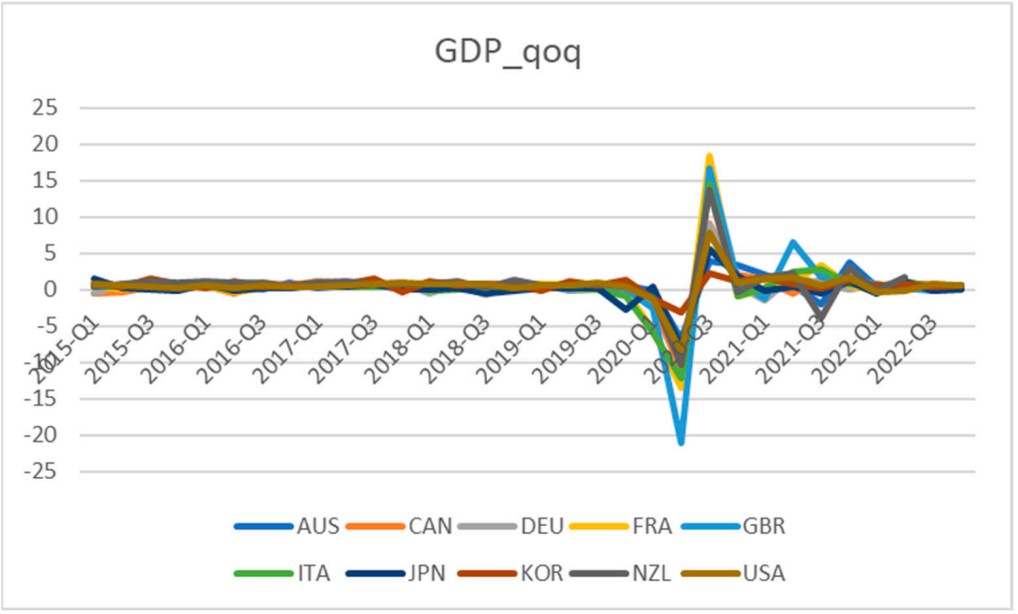

**Figure 4.** Quarter-on-quarter percentage change in the GDPs of ten countries from 2015 Q1 to 2022 Q3. Source: data from OECD (2022a) [36]. Legend: Australia (AUS), Canada (CAN), Germany (DEU), France (FRA), Great Britain (GBR), Italy (ITA), Japan (JAP), South Korea (KOR), New Zealand (NZL), and the United States of America (USA).

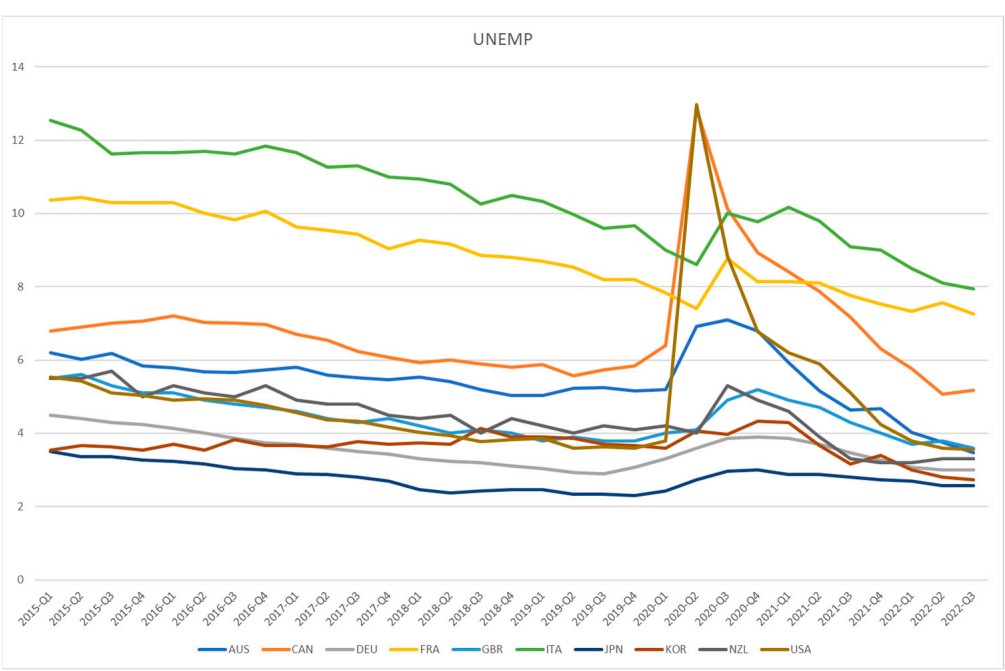

**Figure 5.** Unemployment rates of ten countries from 2015 Q1 to 2022 Q3. Source: data from OECD (2022a) [36]. Legend: Australia (AUS), Canada (CAN), Germany (DEU), France (FRA), Great Britain (GBR), Italy (ITA), Japan (JAP), South Korea (KOR), New Zealand (NZL), and the United States of America (USA).

The use of two interest rate shocks of opposite directions within a short time is one of the major novelties of this study. This is because, even though the rebounds in house prices globally after the outbreak of COVID-19 in 2020 were found empirically to be caused by the synchronized reductions in interest rates and the more accommodative monetary policies of central banks [27], there are some contentions that the pandemic also brought demands for more residential space [58], as well as that the supply chain disruptions during the pandemic could also result in higher construction costs and less housing supply [59]. Previous single-shock quasi-experiments using COVID-19 did not control well for the variable of housing supply.

This study exploited two interest rate shocks, as the supply chain disruptions have not been completely resolved, and the demand for bigger houses is still strong, but house prices in many countries have started leveling off or even falling down. Yiu (2023) [51] explicitly controlled for housing supply in a one-country two-shock study and confirmed the monetary policy hypothesis. This paper further extended to a ten-country two-shock model to tackle the insufficient housing supply contentions. In other words, by including the global inflation crisis as a quasi-experiment, it could provide a strong test for the monetary policy hypothesis, as insufficient housing supply scenarios can only make an underestimation of the global interest rate hike effect.

The concerted efforts of central bankers in cutting interest rates in early 2020 and raising rates in early 2022 triggered two synchronized global shocks for short-term interest rates. However, the long-term interest rate is considered more relevant to housing markets as it is market determined and represents market expectations. As shown in Figures 1 and 6, the patterns of short-term and long-term interest rates of the ten countries in this period were highly similar with some differences. First, they all showed in general a U-shaped (a downward and then an upward) trend. The almost-zero long-term rates of JPN were also similar to the short-term rate, but the long-term rates of ITA were positive and much higher than its short-term rate. The long-term rates of Germany and France also dived into the negative zone during the COVID-19 period.

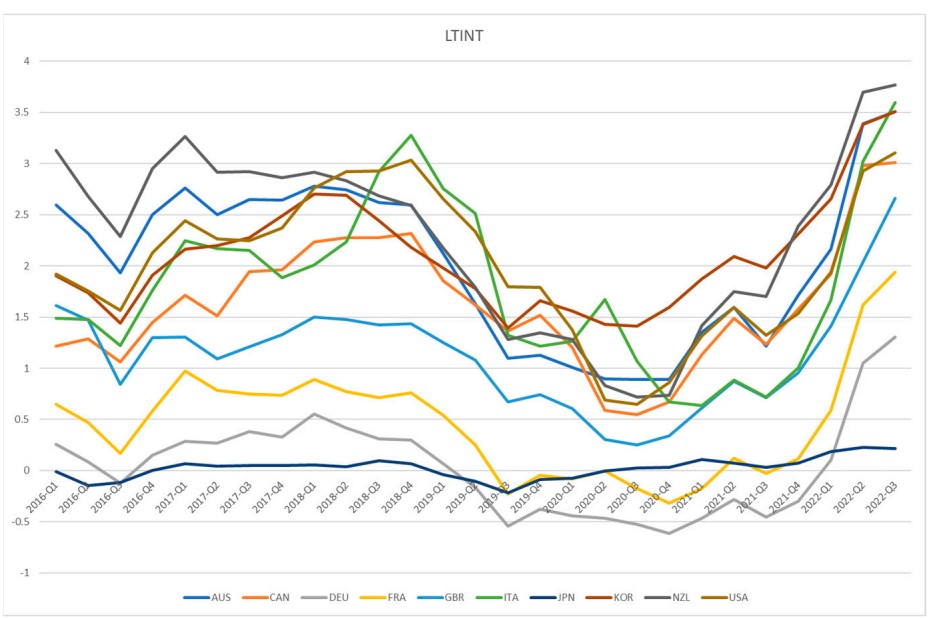

**Figure 6.** Long term interest rates before and after the outbreak of COVID-19. Source: data from OECD (2022a) [36]. Legend: Australia (AUS), Canada (CAN), Germany (DEU), France (FRA), Great Britain (GBR), Italy (ITA), Japan (JAP), South Korea (KOR), New Zealand (NZL), and the United States of America (USA).

## 4. Results

Figure 7 shows the correlation scatterplots of each country from 2016 Q1 to 2022 Q3. A universal negative association between house price changes and real interest rates is found by the best fit lines for all ten countries. However, a scatterplot does not control for other factors, such as the economic growth and unemployment effects. The results of the panel regression analysis are reported below.

The empirical results of the ten-country panel regression model, with cross-country fixed effect and AR (1), are reported in Table 3. In the ten-country full period panel model (Model 1), the sign and magnitude of the quarterly change in the real short-term interest rate d(RSIR) coefficient are negative at about −0.58% and statistically significant at the 1% level, ceteris paribus. Excluding the two control group countries (ITA and JPN) in Model 2, the eight-country panel model shows a stronger negative effect (–0.62%) of real interest rate change on house price change. The control group (Model 3) shows a weak negative but insignificant effect of real interest rate change on house price change. These results reinforce the casual observations that countries with stronger changes in real interest rate impose stronger effects on house price changes.

When dividing the period in two, with only the first global interest rate shock period from 2015 Q1 to 2021 Q3 in Model 4 and both the first and the second shock period from 2018 Q4 to 2022 Q3 in Model 5, the d(RSIR) effect in Model 4 is less negative at about –0.32% and that in Model 5 was the most negative at about –0.71%. Both are significant at the 1% level, *ceteris paribus*. The magnitude indicates a stronger impact of monetary policy change on housing market during the whole pandemic period in Model 5. Further taking both cross-country and period fixed effects, the results in Table 4 also confirm the negative impact of real interest rate on house price change at a similar magnitude.

In this special period, there were two global and synchronized interest rate interventions imposed by central bankers in the wake of the outbreak of the COVID-19 pandemic in 2020 and the emergence of the global inflation since 2022. The synchronized responses of house prices in correspond to the changes in real interest rates in the countries, therefore, implying a causation relationship of real interest rate change on house price change on the basis of temporal precedence and counterfactual dependence.

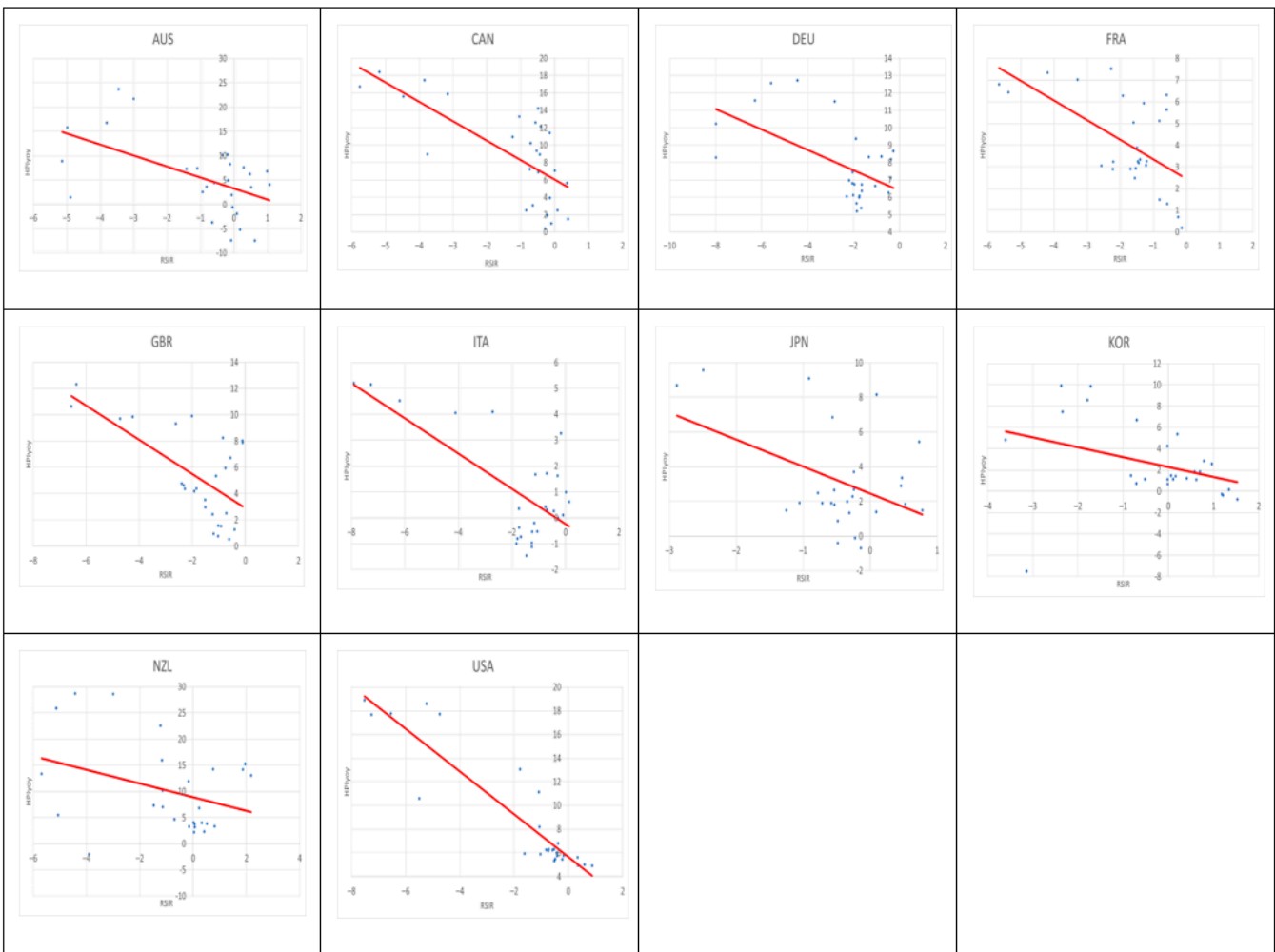

**Figure 7.** Scatterplots of house prices year-on-year changes (HPIyoy) and real short-term interest rates (RSIRs), for 2015 Q1–2022 Q3, in ten countries. Source: data from OECD (2022a) [36]. Legend: Australia (AUS), Canada (CAN), Germany (DEU), France (FRA), Great Britain (GBR), Italy (ITA), Japan (JAP), South Korea (KOR), New Zealand (NZL), and the United States of America (USA).

**Table 3.** Results of the ten-country panel regression models on house price change and real short-term interest rates, with cross-country fixed effect and AR (1).

| Dependent Variables | Model 1—Ten-Country Panel, 2015 Q1–2022 Q3 | Model 2—Eight-Country Panel, 2015 Q1–2022 Q3 | Model 3—Two-Country Panel, 2015 Q1–2022 Q3 | Model 4—Ten-Country Panel, 2015 Q1–2021 Q3 | Model 5—Ten-Country Panel, 2018 Q4–2022 Q3 |
|---|---|---|---|---|---|
| Constant | 0.0112 (6.42) *** | 0.0124 (5.45) *** | 0.0057 (3.01) *** | 0.0142 (7.12) *** | 0.0127 (4.41) *** |
| d(RSIR) | −0.0058 (−4.77) *** | −0.0062 (−4.53) *** | −0.0012 (−0.56) | −0.0032 (−3.23) *** | −0.0071 (−4.17) *** |
| dlog(GDP) | −0.0002 (−0.79) | 0.0001 (0.41) | −0.0009 (−2.54) ** | −0.0001 (−0.85) | −0.0002 (−0.84) |
| d(UNE) | −0.0016 (−1.98) ** | −0.0011 (−1.26) | −0.065 (−1.63) | −0.0017 (−3.02) *** | −0.0016 (−1.61) |
| AR (1) | 0.5573 (9.16) *** | 0.5963 (8.88) *** | 0.4157 (2.96) *** | 0.6878 (13.61) *** | 0.5498 (6.20) *** |
| Dep. Var. | | | dlog(HPI) | | |
| Fixed Effect | | | Cross-country | | |
| No. of Observations | 10 countries × 29 quarters | 8 countries × 29 quarters | 2 countries × 29 quarters | 10 countries × 25 quarters | 10 countries × 16 quarters |
| Adj. R-sq | 0.43 | 0.43 | 0.32 | 0.59 | 0.39 |

Figures in parentheses are t-statistics; *** and ** represent *p*-values ≤ 0.01 and 0.05, respectively.

The economic growth factor shows a negative and insignificant association with house price change in most of the models. The unemployment rate factor shows a negative association

with house price change and is only statistically significant in Model 1 and Model 4. In addition, all models show a strong autoregressive pattern in house price change.

Intuitively, the results showed that housing prices are shaped by monetary policy. An expansionary or contractionary monetary policy triggered a rise or fall of house prices in the world, respectively, after controlling for the economic, unemployment, individual country, and period fixed effects.

**Table 4.** Results of the ten-country panel regression models on house price change and real short-term interest rates, with both cross-country and period fixed effects.

| Dependent Variables | Model 1a—Ten-Country Panel, 2015 Q1–2022 Q3 | Model 2a—Eight-Country Panel, 2015 Q1–2022 Q3 | Model 3a—Two-Country Panel, 2015 Q1–2022 Q3 | Model 4a—Ten-Country Panel, 2015 Q1–2021 Q3 | Model 5a—Ten-Country Panel, 2018 Q4–2022 Q3 |
|---|---|---|---|---|---|
| Constant | 0.0121 (13.74) *** | 0.0143 (13.33) *** | 0.0066 (7.83) *** | 0.0130 (18.55) *** | 0.0135 (9.05) *** |
| d(RSIR) | −0.0053 (−3.03) *** | −0.0037 (−1.66) * | 0.0036 (1.82) * | −0.0015 (−0.95) | −0.0067 (−2.69) *** |
| dlog(GDP) | −0.0002 (−0.37) | −0.0001 (−0.06) | −0.0029 (−3.99) *** | −0.0004 (−0.99) | −0.0002 (−0.27) |
| d(UNE) | −0.0017 (−1.52) | −0.0011 (−0.89) | −0.0002 (−0.06) | −0.0015 (−1.69) * | −0.0019 (−1.34) |
| Dep. Var. | | | dlog(HPI) | | |
| Fixed Effects | | | Cross-country and Period | | |
| No. of Observations | 10 countries × 30 quarters | 8 countries × 30 quarters | 2 countries × 30 quarters | 10 countries × 26 quarters | 10 countries × 16 quarters |
| Adj. R-sq | 0.40 | 0.42 | 0.71 | 0.53 | 0.39 |

Figures in parentheses are t-statistics; *** and * represent *p*-values ≤ 0.01 and 0.10, respectively.

*Robustness Tests*

As a robustness test for the sensitivity of the terms of interest rates, Table 5 shows the empirical results of using real long-term interest rates in the regression models. The results are highly similar to the ones using real short-term interest rates above. For example, in the ten-country panel models (Model 6 and Model 10), the effects of d(RLIR) are negative at about –0.29% and –0.34% on house price changes respectively. The signs, magnitudes and significances of the effects of the GDP growth factor and unemployment rate factor are almost the same as above. The results indicate that the market long-term interest rates are strongly shaped by the central banks' short-term interest rates [60]. Using either short-term or long-term interest rates does not greatly affect much the results of the effect of real interest rate change on house price change.

**Table 5.** Results of the ten-country panel regression models on house price change and real long-term interest rates, with cross-country fixed effect and AR (1).

| Dependent Variables | Model 6—Ten-Country Panel, 2015 Q1–2022 Q3 | Model 7—Eight-Country Panel, 2015 Q1–2022 Q3 | Model 8—Two-Country Panel, 2015 Q1–2022 Q3 | Model 9—Ten-Country Panel, 2015 Q1–2021 Q3 | Model 10—Ten-Country Panel, 2018 Q4–2022 Q3 |
|---|---|---|---|---|---|
| Constant | 0.0114 (5.72) *** | 0.0123 (4.67) *** | 0.0058 (2.99) *** | 0.0142 (6.91) *** | 0.0131 (3.77) *** |
| d(RLIR) | −0.0029 (−2.23) ** | −0.0034 (−2.26) ** | −0.0006 (−0.30) | −0.0027 (−2.71) *** | −0.0034 (−1.82) * |
| dlog(GDP) | −0.0001 (−0.37) | 0.0002 (0.87) | −0.0009 (−2.51) ** | −0.0001 (−0.81) | −0.0001 (−0.40) |
| d(UNE) | −0.0016 (−1.95) * | −0.0010 (−1.14) | −0.0066 (−1.66) | −0.0016 (−2.73) *** | −0.0017 (−1.58) |
| AR (1) | 0.5969 (9.95) *** | 0.6345 (9.55) *** | 0.4386 (3.18) *** | 0.6957 (13.99) *** | 0.6043 (6.91) *** |
| Dep. Var. | | | dlog(HPI) | | |
| Fixed Effect | | | Cross-country | | |
| No. of Observations | 10 countries × 29 quarters | 8 countries × 29 quarters | 2 countries × 29 quarters | 10 countries × 25 quarters | 10 countries × 16 quarters |
| Adj. R-sq | 0.39 | 0.39 | 0.32 | 0.59 | 0.34 |

Figures in parentheses are t-statistics; ***, ** and * represent *p*-values ≤ 0.01, 0.05 and 0.10, respectively.

Another robustness test for the sensitivity of the cross-section heteroscedasticity is conducted by re-estimating the long-term interest rate models with panel EGLS cross-

section weights and White cross-section standard errors and covariance. Table 6 shows the empirical results, which improve the significance and adjusted R-squared values of the results in Table 5.

**Table 6.** Results of the ten-country panel regression models on house price change and real long-term interest rates, with panel EGLS (cross-section weights and White cross-section standard errors and covariance.

| Dependent Variables | Model 6a—Ten-Country Panel, 2015 Q1–2022Q3 | Model 7a—Eight-Country Panel, 2015 Q1–2022 Q3 | Model 8a—Two-Country Panel, 2015 Q1–2022 Q3 | Model 9a—Ten-Country Panel, 2015 Q1–2021 Q3 | Model 10a—Ten-Country Panel, 2018 Q4–2022 Q3 |
|---|---|---|---|---|---|
| Constant | 0.0117 (8.68) *** | 0.0127 (7.80) *** | 0.0058 (3.18) *** | 0.0150 (5.22) *** | 0.0136 (7.05) *** |
| d(RLIR) | −0.0025 (−2.88) *** | −0.0029 (−3.13) *** | −0.0013 (−0.79) | −0.0018 (−2.26) ** | −0.0030 (−2.53) ** |
| dlog(GDP) | −0.0002 (−1.31) | −0.00003 (−0.15) | −0.0011 (−5.40) *** | −0.0002 (−1.02) | −0.0002 (−2.10) ** |
| d(UNE) | −0.0024 (−3.63) *** | −0.0018 (−3.13) *** | −0.0052 (−1.74) * | −0.0017 (−4.97) *** | −0.0028 (−3.51) *** |
| AR (1) | 0.5548 (7.86) *** | 0.5963 (8.37) *** | 0.4407 (2.98) *** | 0.7515 (11.16) *** | 0.5042 (4.86) *** |
| Dep. Var. | | | dlog(HPI) | | |
| Fixed Effect | | | Cross-country | | |
| Homoscedasticity | | Panel EGLS (cross-country weights) and White cross-section standard errors & covariance | | | |
| No. of Observations | 10 countries × 29 quarters | 8 countries × 29 quarters | 2 countries × 29 quarters | 10 countries × 25 quarters | 10 countries × 16 quarters |
| Adj. R-sq | 0.47 | 0.47 | 0.41 | 0.66 | 0.39 |

Figures in parentheses are t-statistics; ***, ** and * represent *p*-values ≤ 0.01, 0.05 and 0.10, respectively.

The third robustness test is to extend the sampled countries to all the OECD countries with RIR and HPI data available, to eliminate any sample selection biases, as the treatment and control groups in the ten-country sample are not randomly assigned. Table 7 shows that the negative effect of real interest rate changes on house price changes is valid and has a similar magnitude in the 36-country panel regression model (Model 11). Several robustness tests are conducted to examine the sensitivity of the level and volatility of real interest rates. Models 12 and 13 show the results of the countries with high and low RIR, and Models 14 and 15 show those of volatile and stable RIR. The former is defined as when the mean of a country's RIR is higher or lower than the mean of the cross-country means, whereas the latter is defined as when the standard deviation of a country's RIR is higher or lower than the mean of the cross-country standard deviations. All of the results show the same signs, with similar magnitudes in the coefficients of the real interest rate changes. They confirm that the monetary policy effect on house prices is valid across all the OECD countries and is not limited to the ten countries studied above.

**Table 7.** Robustness tests results of the 36-country panel regression models on house price change and real long-term interest rates, and sensitivity tests.

| Dependent Variables | Model 11—36-Country Panel, 2015 Q1–2022 Q3 | Model 12—21-Country Panel of Low RIR, 2015 Q1–2022 Q3 | Model 13—15-Country Panel of High RIR, 2015 Q1–2022 Q3 | Model 14—20-Country Panel of Stable RIR, 2015 Q1–2022 Q3 | Model 15—16-Country Panel of Volatile RIR, 2015 Q1–2022 Q3 |
|---|---|---|---|---|---|
| d(RLIR) | −0.0023 (−2.86) *** | −0.0021 (−1.94) * | −0.0025 (−2.00) ** | −0.0020 (−1.65) * | −0.0020 (−1.76) * |
| Dep. Var. | | | dlog(HPI) | | |
| Fixed Effect | | | Cross-country and Period | | |
| No. of Observations | 36 countries × 31 quarters | 21 countries × 31 quarters | 15 countries × 31 quarters | 20 countries × 31 quarters | 16 countries × 31 quarters |
| Adj. R-sq | 0.36 | 0.37 | 0.34 | 0.34 | 0.31 |

Figures in parentheses are t-statistics; ***, ** and * represent *p*-values ≤ 0.01, 0.05, and 0.10, respectively.

## 5. Discussion

Even though housing prices have been escalating to such a level that housing in many cities is considered unaffordable [61], many people are keen to jump on the bandwagon by taking a longer mortgage repayment period or borrowing more by paying a mortgage insurance premium. One of the major reasons why so many people are fond of homeownership is because of the wealth accumulation benefit of housing assets. "Housing is the principle vehicle of wealth accumulation for the typical household is because it can be acquired with debt" [13]. According to OECD (2022b) data [62], housing wealth represents 50.4% of household total wealth on average, and mortgage debt accounts for the largest part (68.6% on average) of total household debt. As a result, housing and monetary policies have wealth distributional implications.

Thus, it is completely imaginable that, when the central banks reduced interest rates sharply and relaxed mortgage loan restrictions at the beginning of the pandemic, investors and home buyers were strongly incentivized to buy houses. On the one hand, lower interest rates and higher loan-to-value ratios imply that home buyers can pay smaller mortgage repayment amounts and invest with higher leverage, i.e., a smaller down-payment. On the other hand, interest rate also determines the market required yield of investments [45], and a lower yield rate prices property higher, keeping rental income constant.

In contrast, when interest rate is increased, homeowners with outstanding mortgage loans and new homebuyers have to pay more for their monthly mortgage payments, as the most common types of mortgage loans in developed countries are either floating rate or short-term fixed rate. In addition, they also have to bear the risk of having their housing assets becoming negative equities (market price falls below the outstanding loan amount). This can incur substantial capital loss if they have to sell the asset, or it may also trigger the loan-call clause in mortgage contracts where lenders can demand that borrowers repay the loans immediately.

Nowadays, housing affordability is more commonly measured by the mortgage repayment amount to income ratio, as it can accurately estimate the proportion of disposable income to be consumed and can be benchmarked with the rental payment for a similar house. However, it depends a lot on monetary policy, including mortgage rate, loan-to-value ratio, debt-to-income ratio, etc. In some countries, families are encouraged to own their homes, and favourable mortgage policies, such as negative gearing and higher loan-to-value ratio for first-time home buyers, etc., are provided. In the past few decades when interest rates had been relatively low, house prices in many countries escalated rapidly, investors and homebuyers were attracted to the housing markets. Housing is considered unaffordable when the house price to income ratio is measured. However, buying homes can become more attractive and affordable if the mortgage repayment amount to income ratio is measured. Many low-to-medium-income households may consider it a golden opportunity to accumulate wealth by acquiring homes with mortgages when borrowing and repayment are easier. Thus, mortgage loans constitute the single largest debt item of households over their lifetimes.

When central bankers increase interest rates and tighten mortgage-lending restrictions, house prices fall but the mortgage repayment amount increases. Is housing more affordable as the house price to income ratio falls? Interestingly, housing becomes even more unaffordable than before for home buyers with credit constraints. Even though house prices have decreased, potential home buyers do not consider housing more affordable even when their incomes are unchanged, as mortgage loans are more expensive and down-payments and market risks are much higher.

Contrary to the proposal of the World Economic Forum [12], it can be a myth to use monetary policy to solve the housing affordability problem. A loosened mortgage policy seems to enable more people to buy homes with smaller down-payments and lower mortgage rates, etc., but it causes higher house prices, which can exacerbate the housing affordability problem. On the contrary, a tightened mortgage policy can bring down house

prices, but it discourages people from buying homes, not just because of the more expensive mortgage loans, but also due to the human instinct to avoid making an expected loss.

In the 2008 global recession and the 2020 COVID-19 pandemic, the global expansionary monetary policy saved many investors and businessmen, or even allowed many new homebuyers to become rich. However, in the 2022 global inflation, the contractionary monetary policy also turned many property investments into negative equities. Many homeowners are in heavy debt and are extremely stressed when repaying higher and higher mortgage repayments. In both situations of expansionary and contractionary monetary policies, housing is considered unaffordable.

## 6. Conclusions

This study makes two contributions. First, this is an attempt to apply two consecutive quasi-experiments with global interest rate interventions in opposite directions to study the monetary policy hypothesis. It confirms the causality relationship from real interest rate on house price change by eliminating any potential endogeneity bias of reverse causality. Second, this study found a synchronized rise and fall of house prices in ten countries in response to the synchronized cutting and raising of interest rates by the central bankers of the countries. It shows the generalizability of the monetary policy theory, both temporally and spatially.

This study exploits two special periods to conduct the quasi-experiments. The first one is the outbreak of the pandemic in 2020 when the global economy was negatively affected. The central banks of many countries implemented synchronized interest rate cuts, and other easing measures, to save the economies. The second period is the emergence of the global inflation crisis in 2022. With the outbreak of the Russian-Ukraine war in February 2022, fuel and food prices were escalated. As a result, inflation rates in many countries were sharply surged. The central banks raised interest rates swiftly and aggressively to fight against the global inflation. These two global synchronized interest rate shocks in opposite directions provided a unique opportunity to empirically test the monetary policy hypothesis in terms of a treatment (expansionary monetary policy) and a treatment reversal (contractionary monetary policy), in countries with different macro- and micro-economic conditions. The universal validity of the hypothesis in the ten countries could eliminate almost all other potential endogeneity issues, such as housing supply and population growth problems. Better still, since a contractionary policy period closely followed an expansionary policy period, the housing supply hypothesis or the population growth hypothesis could hardly be plausible explanations for the house price changes in these periods, as they could not tenably switch from a global insufficient supply in 2020 to a global oversupply in 2022. Furthermore, we control for the effects of economic growth and unemployment rate in the models, and they are found mostly insignificant in the testing periods. This is the first cross-country study exploiting a quasi-experimental approach with both a treatment and a treatment reversal to test the monetary policy hypothesis on house prices. It can help mitigate the endogeneity bias and establish the causality relationship between real interest rate and house prices.

This study, however, is limited by the small number of studied countries and the short period of the time series after the outbreak of the pandemic, as it may take a longer time for some housing markets to reveal the impacts of the contractionary monetary policy since 2022. It requires a further study to cover more countries and a longer period.

As central banks are empowered to have monopolized control over money and monetary policy [63] (Harvey, 2015), the future of the global housing market is inevitably in the hands of central bankers. In the past two decades, especially after the 2008 global financial crisis, when expansionary monetary policies with ultra-low interest rates and quantitative easing were implemented, house prices have skyrocketed which has resulted in public outcries of unaffordable housing and serious wealth inequality issues [13,64]. Now, when central banks have tightened their monetary policies, house prices have started decreasing. The housing affordability problem has not been solved, instead giving many homeowners

heavy debt burdens with the difficulty of paying increasing mortgage repayments, as well as putting off prospective buyers.

Promoting homeownership has been a common policy objective for many governments, and relaxation of down-payment constraints on mortgage loans and tax relief on mortgage debt financing, etc. were found to increase homeownership rates [14]. The main argument for favoring homeownership is the positive externalities for societies, especially in wealth accumulation [13]. It implies that wealth redistributions in the world can largely be shaped by central banks' monetary policies. It probably explains why the housing market can be exploited as an economic growth engine by the financialization of housing and the securitization of residential mortgages [65,66]. It not only supports a bigger development and construction sector, but it also generates huge profits for banking and finance markets, employment opportunities in professional industries, etc. Malpass and Rowlands (2010) [67] even reported that 'governments utilized increasing levels of home ownership as a means of making citizens responsible consumers by exposing them to the outcomes of monetary policy'.

Central banks have their mandates to use monetary policies to achieve price stability (low and stable inflation) and to help manage economic fluctuations. Their impacts on house prices and asset prices are inevitable. However, it may be a mal-prescription to harness monetary policy to attain any goals for housing affordability, homeownership rates, or wealth accumulation, as it can distort housing market dynamics.

**Funding:** This research received no external funding.

**Institutional Review Board Statement:** Not applicable.

**Informed Consent Statement:** Not applicable.

**Data Availability Statement:** All datasets used in this study are publicly available. They can be found at OECD (2022a) [36].

**Conflicts of Interest:** The authors declare no conflict of interest.

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
