# Peer review of "Are Central Banks’ Monetary Policies the Future of Housing Affordability Solutions"

_urbansci, doi:10.3390/urbansci7010018_

Round 1

Reviewer 1 Report

This study conducts a panel regression analysis on ten countries, namely Australia, Canada, France, Germany, Great Britain, Italy, Japan, South Korea, New Zealand, and the United States of America, to test the hypothesis. The results confirm that the change of real interest rate imposes a negative and significant effect on house price growth rates after controlling for economic growth factors, unemployment factors, and cross-country fixed effects.

Although the paper is well written, it is not very informative as to why these countries and variables need to be studied. For me this is a major limitation of the paper.

The authors just mention that some studies do this and other studies do that but they did not explain the reason for study or study contribution/ Study gap is missing, what the previous studies did not incorporate or how the current is unique from existing literature.

The abstract has several shortcomings and lacks several important details even though it is excessively long. For instance, some important details around policy implications would be useful in the abstract.

More importantly, the study lacks a series of robustness checks that can confirm the validity of the findings? It would be important to see some robustness/sensitivity checks beyond what the authors have done.

No justification for focusing on ten countries.

In the Introduction, you need to connect the state of the art to your paper goals. Please follow the literature review by a clear and concise state of the art analysis. This should clearly show the knowledge gaps identified and link them to your paper goals. Please reason both the novelty and the relevance of your paper goals.

In the literature review, a recent and related studies (see, https://doi.org/10.1177/2321022220980541, https://doi.org/10.1016/j.econmod.2016.07.017 ) could be added in the manuscript.

The empirical analysis although interesting, but I would strongly suggest, that the author/s add more intuitive explanation of the findings. Which are the implications for policy makers, researchers, investors, and even the public in general?

The conclusions should be further developed in order to explore the main advantages of the approach herein followed. What are the policy implications? What are the main differences regarding the outcomes obtained with this approach and the other approaches available in the scientific literature?

Finally, the conclusion lacks paragraphs on the limitations of the study as well as on potential future research streams.

Reviewer 2 Report

Dear authors,

Congratulations for your interesting research. I have some suggestions on how to make your text more attractive for wider audience.

Your topic should be settled in a broader discussion on decreasing housing affordability accross countries. It has been found that young generation has minimum change to buy own housing not only because of increased housing prices (See here, for example:  DOI: 10.52950/ES.2021.10.2.003). Housing contributes to poverty and level of deprivation and is an important topic even in countries experiencing decrease in the overal poverty level (see here: DOI: 10.52950/ES.2022.11.1.009). I suggest referencing these examples from other countries. 

2. It is advisable to reveal the importance of increased housing costs related to increased energy costs. In some countries this share of housing costs makes housing unaffordable even for households having their homes in own possession. This is a resonant issue in Europe currently (see here: https://doi.org/10.3390/en15041281) but perhaps in other countries. This aspect should be mentioned.

3. What I miss in this paper is a policy debate. Should the support (if any…) be aimed at supply or demand side of the market? Social housing projects may increase prices of commercial housing and crowd-out those who are not included in social housing projects. This is a resonant topic for housing policy formulation (see here: DOI: 10.52950/ES.2022.11.2.002).

These are interesting points of view to be included and referenced in your research in order to attract more attention from international audience

 Good luck!

Reviewer 3 Report

I would like to thank the editor for inviting me to review the paper that investigates the impacts of real interest rate on house prices. My major concern focuses on the contribution of this study. The contribution of this paper, as the author claimed, is the elimination of potential endogeneity biases when examining the impact of real interest rate on house prices. Specifically, this study defines the treatment group as those which followed a huge increase in interest rate after the outbreak of the COVID-19 pandemic in 2020 and a huge decrease in interest rate after the outbreak of the Russian-Ukraine war in 2022. The countries which did not follow the interest rate shocks are defined as the control group. This research design obviously indicates that the treatment and control groups are not exogenously assigned. 

In addition, even though the assignment is randomly assigned, I cannot find how the current research design exploits the so-called global interest rate shocks as quasi-experiments. The empirical results, as shown in Table 3, only show the association between real short-term interest rate and house price change is significant in the full sample and treatment group, while insignificant in the control group. The insignificance of real short-term interest rate in the control group is as expected as the control group is defined as those with small changes in interest rate. However, this evidence seems to be irrelevant to the COVID-19 pandemic and the Russian-Ukraine war.

Finally, I am surprised to find that the panel regression model does not control for the time fixed effects, although the country fixed effects are controlled. Overall, the relationship between real interest rate and housing prices has been substantially examined, and I fail to find the marginal contribution of this study.

Round 2

Reviewer 1 Report

The authors have successfully considered and addressed all of my comments made on an earlier draft of this work.

Reviewer 3 Report

Again, I fail to find the marginal contribution of this study theoretically and empirically as the relationship between real interest rate and housing prices has been substantially examined. The assignment of treatment and control groups is far from random, which threats the validity of empirical findings.
